:¤: PLOS | ONE

# *Ex vivo* perfusion-based engraftment of genetically engineered cell sensors into transplantable organs

**Ling-Yee Chin[1,2☯], Cailah Carroll[1,2☯], Siavash Raigani[1,3], Danielle M. Detelich[1,3], Shannon N. Tessier[1,2], Gregory R. Wojtkiewicz[4], Stephen P. Schmidt[4], Ralph Weissleder[4], Heidi Yeh[1,3], Korkut Uygun[1,2], Biju Parekkadan** [1,2,3,5,6]*

**1** Center for Surgery, Innovation, and Bioengineering, Department of Surgery, Massachusetts General Hospital, Boston, Massachusetts, United States of America, **2** Shriners Hospitals for Children, Boston, Massachusetts, United States of America, **3** Center for Transplant Sciences, Massachusetts General Hospital, Boston, Massachusetts, United States of America, **4** Center for Systems Biology, Massachusetts General Hospital, Boston, Massachusetts, United States of America, **5** Harvard Stem Cell Institute, Cambridge, Massachusetts, United States of America, **6** Department of Biomedical Engineering, Rutgers University, Piscataway, New Jersey, United States of America

☯ These authors contributed equally to this work.
\* biju.parekkadan@rutgers.edu

**Data Availability Statement:** All relevant data are within the manuscript and its Supporting Information files.

## Abstract

Cellular rejection of liver transplant allografts remains a concern despite immunosuppressant use. Existing transplant biomarkers are often not sensitive enough to detect acute or chronic rejection at an early enough stage to allow successful clinical intervention. We herein developed a cell-based sensor that can potentially be used for monitoring local events following liver transplantation. Utilizing a machine perfusion system as a platform to engraft the cells into a donor liver, we effectively established the biocompatibility of the biosensor cells and confirmed efficient delivery of cells distributed throughout the organ. This work proves an innovative concept of integrating synthetic reporter cells *ex vivo* into organs as a <u>transplant-within-a-transplant</u> during functional organ preservation with a vision to use cell biosensors as a broad way to monitor and treat tissue transplants.

## Introduction

Liver transplantation remains the definitive surgical cure for end-stage liver disease. The last several decades have seen significant advances in the safety of transplant surgery and outcomes of recipients. Advances in immunosuppression, including the introduction of calcineurin inhibitors and mammalian target of rapamycin (mTOR) inhibitors, have improved rejection rates [1]. Detection of graft rejection, however, has lagged behind these developments. Diagnosis of post-transplant liver dysfunction, including ischemia-reperfusion injury, acute and chronic rejection, and even steatohepatitis, often requires core liver biopsies (percutaneous or endovascular), often in serial fashion. which is not without risk especially in patients with liver failure. Current serum biomarkers of graft dysfunction are also largely limited to nonspecific

**Funding:** This research was conducted with support under Grant No. R21AI134116 (to BP and KU) awarded by the National Institutes of Health and support from the Shriners Foundation for Children Grant No. 85130 (to BP).

**Competing interests:** A patent on engineered cell technology and uses in organ transplantation entitled "Personalized and time release of biomolecules" has been filed. The authors declare no other competing financial interests. This does not alter our adherence to PLOS ONE policies on sharing data and materials.

liver function tests. Therefore, there remains a barrier to the management of liver transplant patients at the point-of-care. Furthermore, the need for continued monitoring after diagnosis of graft dysfunction is critical given the scarcity of an organ transplants and the costs of organ failure.

Cell therapy to modulate organ dysfunction after transplantation is being increasingly explored for treatment of ischemic-reperfusion injury, prevention of chronic allograft dysfunction, minimization of immune suppression, and induction of long-term allograft tolerance. Many cell types have been investigated as potential cell-based immunotherapies for use in solid-organ transplant, including mesenchymal stromal cells, regulatory macrophages, tolerogenic dendritic cells, regulatory T cells, and regulatory B cells [2–11]. Moreover, the use of concomitant kidney and bone marrow transplants to induce mixed chimerism and tolerance [12, 13] has been explored with initial success. These cell therapies are often administered intravenously with limited half-life in the body [14, 15] and non-specific targeting to an organ bed where modulation or tolerance is needed. Thus, a significant barrier to the use of cell therapeutics to modulate organ recovery after transplant may be an inefficient delivery to sites of pathology.

To overcome the limited half-life *in vivo* and non-specific delivery of cell therapies for transplant modulation applications, we engineered cells to be directly engrafted into an organ prior to transplantation with machine perfusion. A rat fibroblast line was initially chosen for this study. The rationale for selection included the availability and ease of transduction, ability to engraft, and potential use in eventually modulating tissue dysfunction [16]. We did not use mesenchymal stem cells, despite the potential for eventual clinical use, to avoid potential therapeutic effects they may have which would be confounding factors in assessment of the liver function/viability. The scope of this initial work was therefore to establish the integrity of biosensor cells infused into an organ using a constitutive CMV promoter to drive the secretion of *Gaussia* luciferase (gLuc), a bioluminescent biomarker probe [17]. We have previously investigated the pharmacokinetics of a cell therapy coupled with gLuc monitoring of cellular transplant [18] and used this technique to confirm immune clearance of such biomarker-secreting cells [19]. Furthermore, we tested the ability of a previously established *ex vivo* liver perfusion system [20] as a novel and enabling platform for engrafting cell biosensors into the organs prior to transplant. Herein, we describe the process development to verify the successful engraftment of biosensor cells in donor livers, with a robust blood-based biomarker signal and minimal impact on the organ.

## Methods

### Rat fibroblast culture and expansion

Frozen vials of Rat2 fibroblast cell line were purchased from American Type Culture Collection (Manassas, VA, USA). Cells were thawed and cultured in Dulbecco Modified Eagle Medium (DMEM) composed of 10% fetal bovine serum (FBS) and 2% penicillin and streptomycin. Media was changed every 3–4 days and incubated at 37°C, 5% carbon dioxide. Cells were subcultured when they reached 80–90% confluence.

### Genetic engineering of rat fibroblasts

Rat fibroblasts were harvested at passage 2 for lentiviral infection. A lentivirus vector expressing gLuc [17, 21] and green fluorescent protein (GFP) under the control of the CMV promoter was obtained from the Massachusetts General Hospital Vector Core (funded by NIH/NINDS P30NS045776). Cells were cultured for 24h in DMEM with increasing concentrations of lentiviral particles per cell and protamine sulfate, a cationic vehicle [22]. Transduced GFP-positive

cells were sorted using a BD FACS Aria III (BD Biosciences) cell sorter (Harvard Stem Cell Institute Flow Cytometry Core at Massachusetts General Hospital, Boston, MA, USA). GFP-positive cells were then cultured, expanded and used for subsequent studies. Only passages 3–5 rat fibroblasts were used for experiments.

### Animals

Male Lewis rats weighing 200g-250g were housed in standard conditions (Charles River Laboratories, Boston, MA, USA). The animals were kept in accordance with the National Research Council guidelines. The experimental protocol was approved by the Institutional Animal Care and Use Committee, Massachusetts General Hospital.

### Liver procurement

All procurements were performed using the technique of Delriviere et al [23]. Animals were anesthetized using inhalation of 3–5% isoflurane (Forane, Deerfield, IL, USA) with 1 L/min 95%/5% oxygen-carbon dioxide gas. The animal's abdomen was shaved and a transverse laparotomy was made. Intestines were moved to expose the entirety of the liver, portal vein, common bile duct, and inferior vena cava. The common bile duct was cannulated using a ~6cm 28-gauge polyethylene tube (Surflo, Terumo, Somerset, NJ, USA) to collect bile throughout the perfusion. Via the infrahepatic vena cava (IHVC), 300U of heparin was administered and 3 minutes were allowed for circulation. The portal vein was cannulated using a 16-gauge catheter and IHVC was transected for exsanguination. All cannulas were secured with 7–0 silk suture. The liver was immediately flushed in situ via the portal vein cannula with 50mL of 0.9% NaCl at 4°C. The liver was freed from its ligamentous attachments, weighed, and placed in ice-cold saline prior to being connected to the perfusion circuit.

### Perfusate composition

Perfusate composition consisted of a base of DMEM supplemented with 200mM L-glutamine (Invitrogen), 10% v/v FBS (Thermo Scientific), 5% with bovine serum albumin (BSA; Sigma-Aldrich), 8mg/L dexamethasone (Sigma-Aldrich), 2000 U/L heparin (APP Pharmaceuticals), and 2 U/L insulin (Humulin, Eli Lily).

A determined concentration of $5 \times 10^6$ engineered rat fibroblasts was added to 150mL of perfusate to circulate through the system for the initial three hours. At hour three, the perfusate was switched to media without any rat fibroblasts and perfused for an additional three hours.

### Normothermic machine perfusion

Normothermic machine perfusion (NMP) was chosen to maintain the liver at a metabolically active state, similar to *in vivo* conditions [24, 25]. The NMP circuit used was comprised of an organ reservoir, bubble trap, membrane oxygenator, roller pump, water bath, and series of silicon tubing. Prior to liver procurement, the perfusion system was first flushed with ultrapure water and warmed to 37C before perfusate was circulated.

Immediately after procurement, the liver was transferred to the organ reservoir and perfused through the portal vein cannula. The liver was perfused with partial oxygen pressure ($pO_2$) above 400mmHg. The flow rate of the system was manually altered according to target a portal pressure of 5 mmHg, measured using a water column manometer. Flow rates initiated at 5mL/min and were increased to maintain an absolute pressure of 5 mmHg inside the liver.

## Perfusate analysis

Samples of perfusate were taken at time points 0, 30, 60, 120, 180, 210, 240, 300, and 360 minutes and stored at -80°C. Serum chemistry and blood gas analyses were performed during perfusion using CG4+ and CHEM8+ i-STAT cartridges (Abbott Point of Care Inc., Princeton, NJ, USA). Liver biopsies were taken post-perfusion and either snap-frozen in liquid nitrogen or fixed in 10% formalin. Assays for aspartate aminotransferase (AST; TR70121, Thermo Scientific) were performed following perfusion according to the manufacturer's instructions.

## Bioluminescence assays

Bioluminescence assays were performed by pipetting 10 μL of sample into a black-walled 96-well plate (Corning) and adding 1000 μL of coelenterazine native substrate (NanoLight Technology) at 100 μmol/L diluted in phosphate buffered saline. Samples were read immediately using a BioTek Synergy 2 Multi- Mode Reader for 10s at a gain of 100–200 (BioTek).

## *Ex vivo* liver imaging

Prior to perfusion, $5 \times 10^6$ Rat2 fibroblasts were labeled with a near infrared fluorescent membrane dye (Qtracker 705 Cell Labeling Kit, Invitrogen). *Ex vivo* fluorescence imaging with the 665 excitation and 680 emission filter set was performed using the Olympus OV110 (Olympus Corporation) to visualize engraftment in liver. Image visualizations were performed in ImageJ software.

## Cell lysis of gLuc-secreting cells in liver tissue

Tissue biopsied post perfusion and stored in -80°C was lysed for *Gaussia* Luciferase activity using NanoFuel FLASH Assay for *Gaussia* Luciferase (NanoLight Technology, 319). Approximately 100mg rat liver tissue was homogenized in 200uL lysis buffer, the samples were vortexed and kept on ice. After 15 minutes, 200uL of *Gaussia* dilution buffer was added to the samples and they were vortexed once more. *Gaussia*-expressing cells were used as a positive control and were prepared in the same sequence. Tissue perfused with non-transduced cells was used as a negative control. The sample volume for each well was 20uL with N = 4. Finally, 50uL of Coelenterazine buffer was injected into each well and read for Luminescence with an integration time of 10 seconds.

## Histological evaluation

Once perfusion was completed tissue samples were taken from three different lobe locations. The collected samples were formalin-fixed to be paraffin-embedded. The tissue was cut into 4-micrometer sections, mounted on a glass slide, and stained for hematoxylin-eosin and/or anti-GFP antibody (Abcam ab1218).

## Results

### Optimization and characterization of engineered biosensor cells

We utilized self-inactivating lentiviral vectors as previously described [26] to integrate transgenes into the genome of dividing as well as non-dividing cells and pass them onto daughter cells; hence, cells will be stably expressing all genetic reporters. The lentivirus was engineered to contain a GFP gene for purifying engineered cells by FACS. In addition, a secreted gLuc reporter gene was inserted. gLuc enzyme activity can be specifically and easily quantified in a small aliquot of volume (5 μl) *ex vivo* by adding its respective substrate and measuring

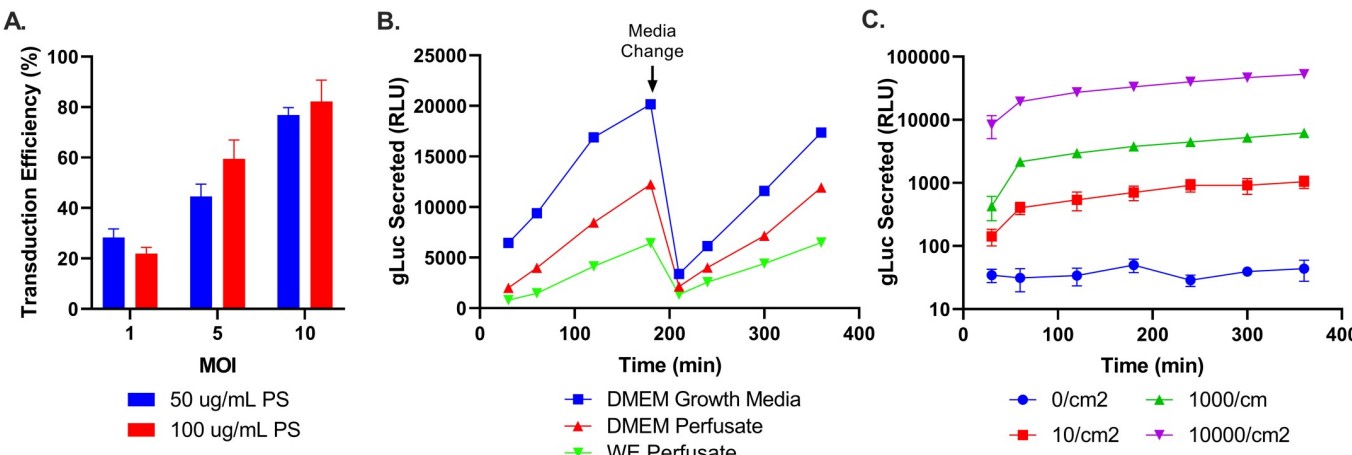

**Fig 1. Characterization and optimization of secreted luciferase reporter system in engineered rat fibroblasts.** (A) Transduction efficiency measured by GFP expression using flow cytometry of Rat2 cells cultured for 24h in DMEM with different lentiviral particles per cell (MOI) or protamine sulfate (PS) concentrations. (B) In vitro gLuc secretion under different media conditions. A media exchange was performed to wash out accumulated gLuc and fresh media was provided to detect continued secretion. A DMEM base resulted in higher gLuc secretion than the previously-used perfusate with a Williams E base. (C) Release of gLuc under different cell seeding densities. Secretion of engineered rat fibroblasts was stable and increase linearly with seeding density in vitro.

enzymatic conversion. gLuc has been used for high sensitivity detection [17, 21] with a half-life of 5–10 minutes in mouse circulation and from as few ~1000 cells in an entire mouse [26, 27]. Since these probes are secreted, accumulate in the blood, and are specific to their corresponding substrates, the signal intensity is very specific and amplified such that even a few dispersed cells can be detected using a simple bioluminescent blood test. This reporter system could enable the indirect observation, in real-time, of cell fate within an organ transplant *in vivo* by measuring biomarker levels in circulating fluids.

Conditions with high concentrations of lentiviral particle MOI and the cationic vehicle had the highest transduction efficiency (Fig 1A). Once rat fibroblasts were transduced with lentiviral particles, gLuc secretion was characterized during a 6-hour in vitro experiment modeled on the planned perfusion with a washout period (Fig 1B). As expected, gLuc secretion rose after cell seeding, was reduced after media exchange, and rose again with a similar rate in fresh media. When transitioning from an *in vitro* to an *ex vivo* organ perfusion model, the typical organ perfusate solution would be the vehicle that engineered cells would be circulating in. We explored gLuc release in vitro between typical fibroblast growth media (DMEM), perfusate using a DMEM base, and perfusate using a Williams E base (as previously utilized for liver perfusion). Perfusate utilizing a DMEM base resulted in higher gLuc secretion (Fig 1B) and was used for subsequent perfusions. The production of gLuc was stable, independent of cell density, and was more sensitive to the concentration of cells per $cm^2$ (Fig 1C) when plotted on a log scale.

## Biosensor cells successfully engraft in *ex vivo* rat livers by machine perfusion

A six-hour perfusion was previously established as a benchmark for rat liver normothermic perfusion (NMP) to preserve liver metabolic function before successful transplant into recipient rats [24]. Given the eventual aim of these biosensor cells to monitor and regulate transplanted livers, we chose to use the same six-hour perfusion in our experiments. In order to confirm successful engraftment of the biosensor cells into the liver, we infused the cells for the first three hours of the perfusion, then used fresh perfusate to check if the cells washed out. An

initial cell mass of 5 x 10$^6$ engineered rat fibroblasts was chosen at a concentration of approximately 30 x 10$^3$ cells/mL to have a dilute cell suspension well below physiological circulating cell numbers.

To confirm the location of the biosensor cells within the liver, they were treated with a near-infrared (NIR) dye prior to perfusion. Cells injected directly through the cannulated liver did not distribute as thoroughly as perfused cells (S1 Fig), justifying the need for a continuous flow-based seeding process for improved biodistribution of engineered cells within the tissue. As the length of perfusion increased from 2 hours (Fig 2A) to 4 hours (Fig 2B) to 6 hours (Fig 2C), we found that longer perfusions corresponded with qualitatively improved distribution of the cells across the organ bed even after a washout period and addition of fresh perfusate. The 6-hour perfusion time qualitatively enabled distribution of cells into the periphery, which was not clearly populated in livers perfused for 2–4 hours.

### Histological confirmation of biosensor cell engraftment

NIR tracking of engineered cells helped initially verify cell engraftment, though could not resolve microscopic resolution of cell localization within the tissue. Histological analysis was further performed to assess the presence of the biosensor cells embedded in the tissue by staining GFP+ cells using for anti-GFP antibody 9F9.F9 (Fig 3A and 3B). Engineered cells contained both GFP and gLuc genes and therefore the location of the cells can be tracked via GFP presence as well. The brown staining in tissue samples indicated GFP+ cells present in the tissue. Cell localization was generally near sinusoidal endothelium near portal veins. Hematoxylin-eosin (H&E) staining was used initially to histologically assess preservation injury and endothelial cell damage in liver tissue. Confluence of cells and absence of neutrophils is an indication of little to no damage caused due to the perfusion and/or addition of cells [28]. Biopsies (Fig 3C and 3D) from different liver perfusions were shown to have no major pathological damage based on these criteria.

### Engrafted biosensor cells have minimal impact on liver viability functionality

Liver functionality after cell engraftment was further assessed at the organ level by comparing the biochemistries of an experimental group perfused with transduced cells, a group

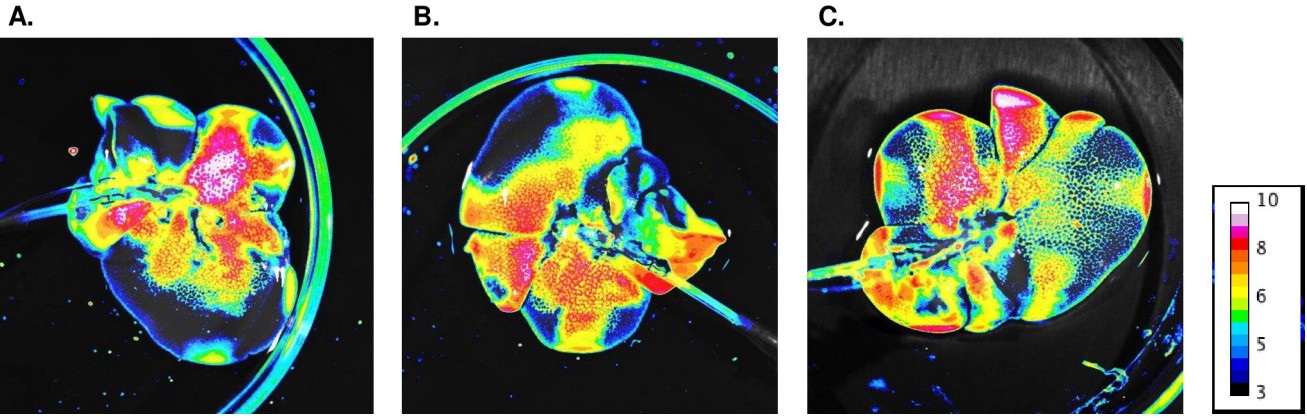

**Fig 2. Cells successfully engrafted into livers and longer perfusion times correlated with increased distribution in the vasculature.** Near-infrared imaging of cells after infusion into livers under (A) 2-hour total perfusion, (B) 4-hour total perfusion, (C) 6-hour total perfusion. Pseudocolor indicates arbitrary intensity of cells in a given region with red-to-blue corresponding to highest-to-lowest intensity.

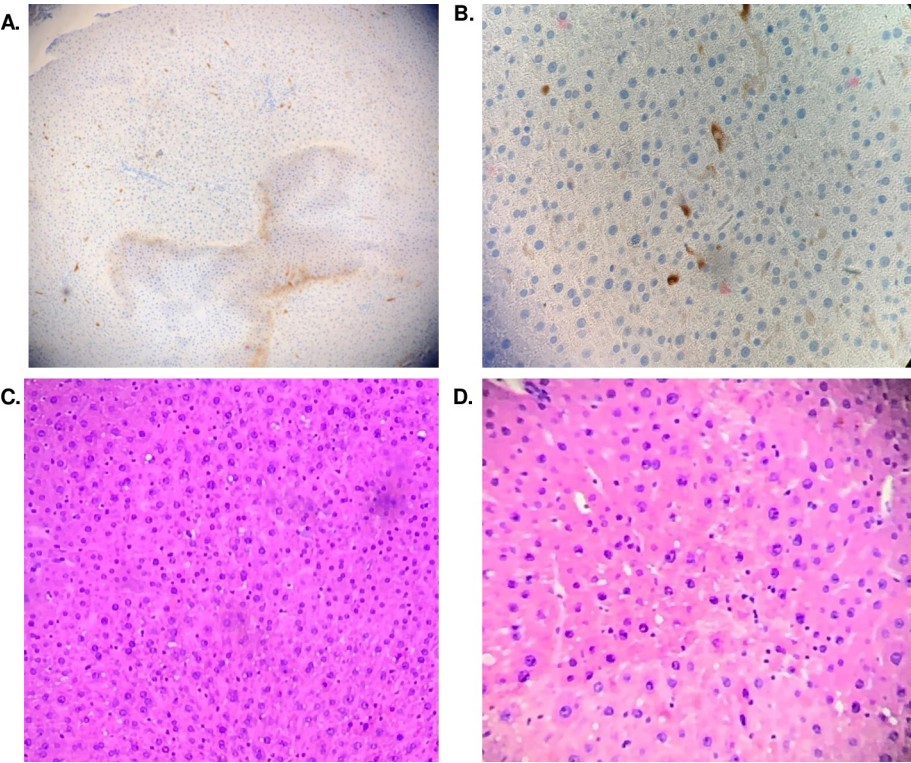

**Fig 3. Histology of liver tissue after cell infusion.** Paraffin embedded liver tissue samples taken post perfusion and stained using an Anti-GFP antibody for the presence of GFP in the tissue stained brown at (A) 10x or (B) 40x magnification. Hematoxylin and eosin stain of paraffin embedded liver tissue samples taken post perfusion to check for endothelial cell damage cause by perfusion or the addition of biosensor cells at (C) 10x magnification or (D) 40x magnification.

containing un-transduced cells, and a negative control group perfused with no cells. These groups could isolate the effect of cells alone compared to the transgene to identify a root cause of any observed effects.

To assess viability of the perfused grafts, we used the composite viability criterion of Mergental et al which were established to clinically predict primary nonfunction in normothermically perfused human livers [29]. Briefly, one major and two minor criteria must be met in order to deem the liver clinically viable. As shown in Fig 4, perfused livers containing engineered sensor cells produce bile (Fig 4A), a major criterion; show stable flow rates (Fig 4B) and stable perfusate pH (Fig 4C), two minor criteria. Therefore, it can be suggested that the cell infusion and engraftment would not have an impact on the ability of the liver to successfully survive and function after transplant.

To further assess if the cell infusion did affect function, we assessed several additional parameters during perfusion: Metabolic activity of the liver was additionally tested using glucose stability [23] as well as oxygen consumption rate. Hepatocellular damage was further assessed using and aspartate transaminase (AST) release [30] which we have also shown to be correlated to transplant success in similar normothermic rat liver perfusions systems [31]. Finally, we measured liver weight before and after perfusion to evaluate if there was any edema. In all three groups lactate (Fig 5A) and glucose (Fig 5B) values rose initially but began to stabilize after the perfusate switch at T = 3hrs. Oxygen consumption (Fig 5C) was calculated using $O_2 \ Consumption = portal \ flow * S * \frac{pO_{2inflow} - pO_{2outflow}}{g \ liver \ weight}$ (S, the solubility constant of water at

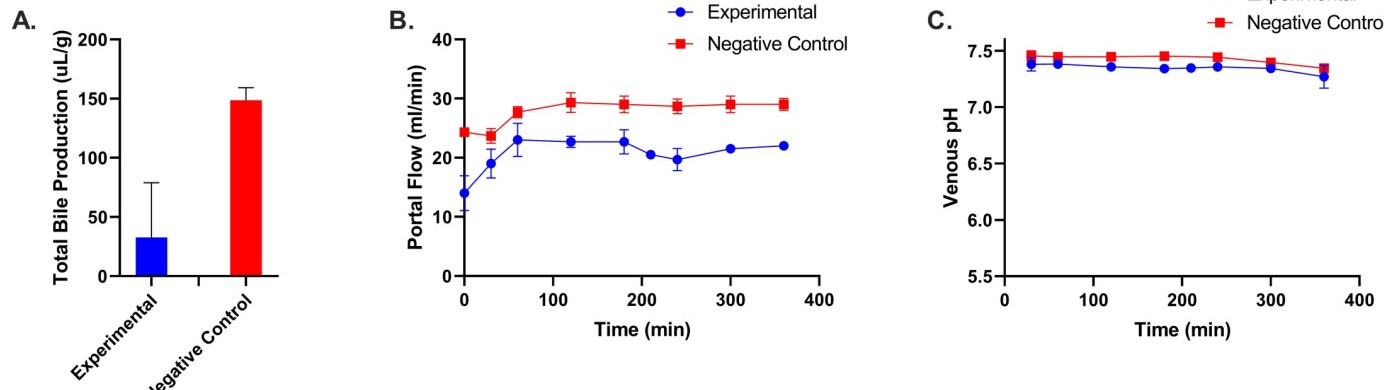

**Fig 4. Transplant viability assessment for biosensor engrafted livers.** (A) Total bile collected during perfusion. (B) Portal flow rates. (C) Venous pH obtained from perfusate.

$37°C = 0.031 μL/mL/mm Hg$) [15]. Oxygen consumption was found to be higher in the control group due to the higher portal flows. AST production, known to correlate with liver damage, followed a similar trend for all three groups throughout perfusion (Fig 5D). Finally, weight change (Fig 5E) pre- and post-perfusion showed that the engineered fibroblasts did not cause any edema to the liver. Overall, these assays show conclusively that the engrafted biosensor cells did not any negative impact on liver function.

## Biosensor cells remain viable and secrete detectable gLuc in external circulation fluids

Biosensor cells were added to perfusate to circulate and engraft in liver, and fresh perfusate without cells was swapped in after 180 minutes to test for engraftment. gLuc has been used as a highly sensitive reporter for assessment of cells *in vivo*, so levels in the perfusate allow us to track its activity within the rat livers [18, 19, 21]. There was a consistent pattern of gLuc secretion: an initial buildup in the first three hours, then a drop at the time of switching to fresh perfusate, and resumption of increase during hours 4–6. (Fig 6A). Given that the rate of production was similar after the washout period, the data suggest that the initial release of gLuc during the engraftment period represented the adherent, engineered cells within the organ itself rather than being derived from circulating cells.

In order to approximate the number of engrafted cells per tissue, we measured intra-tissue gLuc levels from frozen tissue biopsies compared to a known control sample of pure engineered gLuc fibroblasts. Tissue samples were lysed and the transduced cell group, in comparison to the un-transduced cell group (Negative Control), had significantly higher levels of gLuc secretion (Fig 6B). When compared to a positive control of $1 \times 10^6$ pure gLuc cells, the signal coming from a 100mg tissue biopsy was ~13% to that of the positive control. This equates to ~$1.3 \times 10^3$ engineered cells per mg of tissue. When scaled to a 10g liver, the order of magnitude of engineered cells approaches $10^7$ cells which suggests that nearly the entirety of the infused cell population likely engrafted, albeit greater than expected and likely due to the variability of the assay and assumptions herein.

## Discussion

Cell therapy is now an established class of medical therapy with approved products such as chimeric antigen receptor (CAR)-T cells and mesenchymal stromal cells (MSCs) being approved

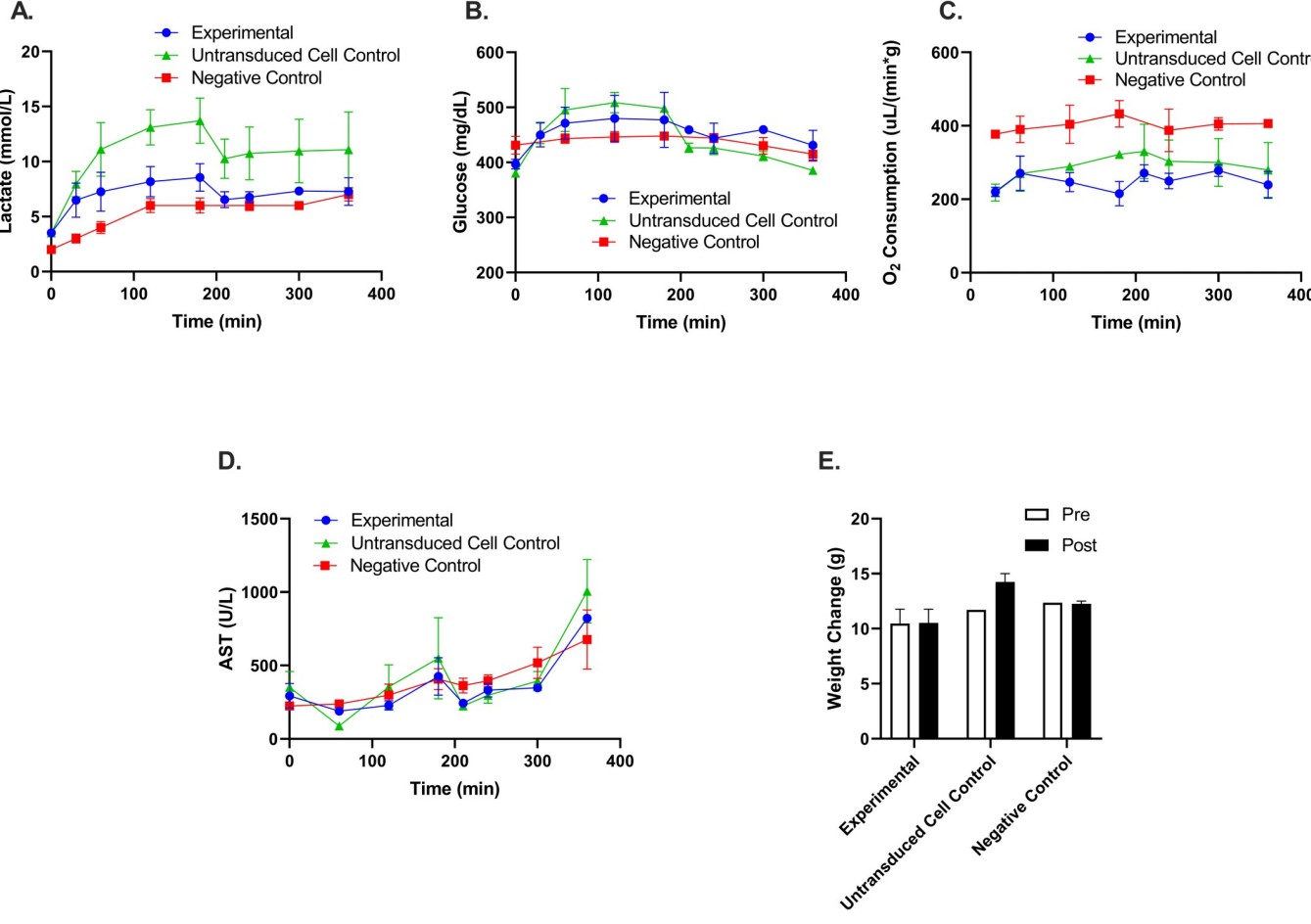

**Fig 5. Engrafted biosensor cells have minimal impact on liver functionality.** (A) Lactate, (B) Glucose levels, (C) Oxygen consumption, and (D) AST levels obtained from perfusate samples collected throughout perfusion. Lactate, glucose, and oxygen were measured in real time using a blood chemistry analyzer (i-STAT). (E) Weight of liver measured pre and post perfusion.

in several jurisdictions around the world. We herein describe the foundation of a strategy to robustly distribute a cell therapy into an organ prior to theoretic transplantation using *ex vivo* machine perfusion. Our approach using machine perfusion has the advantage of efficiently delivering engineered cells into a designated organ. Simple injection of cells distributed poorly (S1 Fig) and therefore may not be able to represent or cover the entirety of the organ with local diffusion being the major form of molecular transport within a tissue bed. At the clinical level, the introduction of machine perfusion into the transplant surgeons' workflow has already decreased organ discard rates [32] and is quickly being widely adopted throughout the world. Therefore, machine perfusion is a unique platform for applying therapeutic adjuncts, including cell therapy, to modulate organ function. In this study, we utilized our previously established *ex vivo* liver perfusion system [20] as a novel platform for engrafting cell biosensors in a pre-transplant model. In confirming the biocompatibility of these cells through liver functionality assays and confirming the efficient delivery of these cells via a normothermic perfusion system, we have developed an efficient platform to further the applications in a transplant dysfunction context. Moreover, since the engraftment of cells does not appear to damage the organ, local cell sensors can also provide a way of monitoring organ quality as a release criteria prior to transplant for the resuscitation of marginal donor organs.

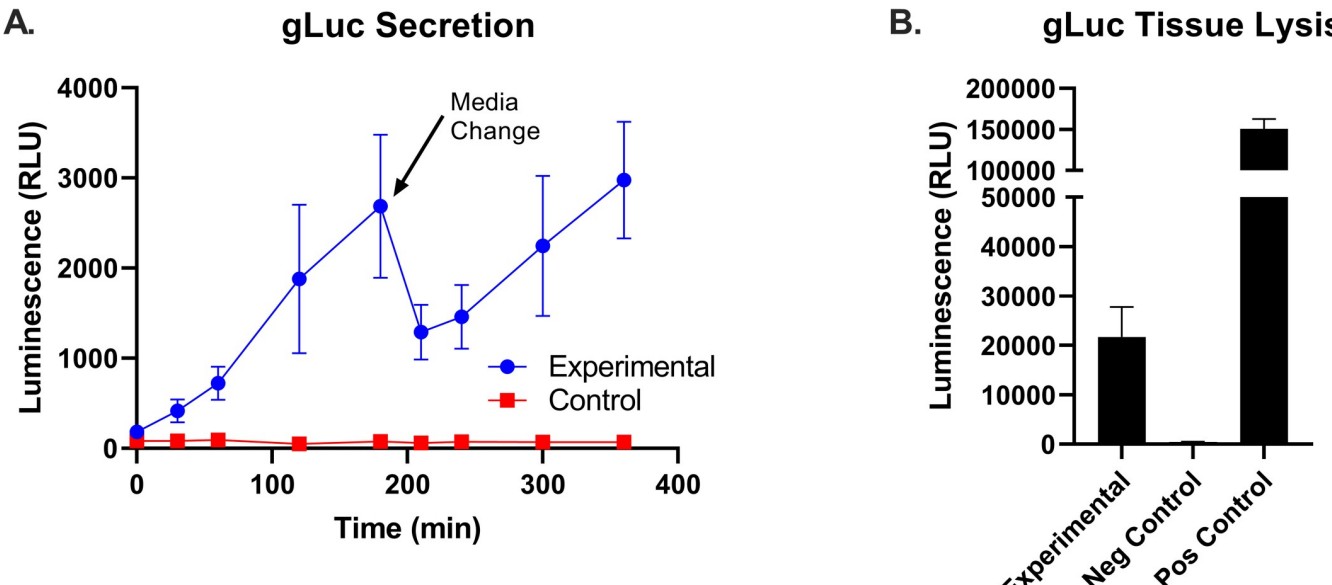

**Fig 6. Release of engineered biomarker from engrafted cells during machine perfusion.** (A) Biosensor cells were infused in perfused livers for the first three hours, then swapped out for fresh perfusate to test engraftment. gLuc was measured in the collected perfusate. Livers displayed a pattern of gLuc secretion in the perfusate consistent with the experiment design. (B) Detection of gLuc from frozen tissue biopsies taken post-perfusion. Tissue lysis for gLuc demonstrates successful engraftment of biosensor cells in the liver tissue.

The perfusion system allows for the engraftment of engineered cells into an organ under stable operating conditions for machine perfusion. The time of perfusion for optimal engraftment and minimal *ex vivo* storage is an area of process development worth improving. Extended perfusion time seems to improve cell distribution to peripheral areas of the organ which can be important to provide comprehensive coverage in the tissue bed to detect local microenvironmental signaling that may only diffuse on the order of 0.1–1 mm in interstitial distance. Varying cell numbers is also worth exploration to evaluate the minimal number of cells necessary to provide local microenvironmental coverage across an organ to impact physiological outcomes. For this purpose, it will be useful to have a diseased organ to use as a testbed in order to find these functional limits. Our initial estimates suggest that $5 \times 10^6$ engineered rat fibroblasts provided a good coverage of cell distribution with near total engraftment. An engrafted population of $\sim 10^3$ cells per mg tissue would ensure good sampling of the organ. More detailed analysis of counting engrafted cells can give further precision to this number and can be a quantitative metric to compare against tissue functional outcome data.

Fibroblasts were tested as a first model cell type due to their ease of isolation from allogeneic sources, efficient transduction, and the opportunity to create well-characterized frozen banks for "off-the-shelf" use given the unpredictable timing of transplant surgery. Fibroblasts durably engrafted into the liver and showed robust, engineered protein secretion. Notably, bone marrow derived fibroblastic MSCs are already in widespread use clinically [33], and genetically engineered MSCs as a therapeutic are already in clinical trials [34] easing the regulatory pathway for future purposes. MSCs have also been demonstrated as a therapeutic adjuvant in perfused human kidneys to induce a regeneration response [35]; it is therefore possible to envision future approaches where such innate therapeutic capabilities are combined with sensor functions engineered into the cells. For this initial study, we did not choose MSCs because such existing therapeutic effects would be possible confounding factors when analyzing the viability and function of livers during perfusion.

This study develops the protocols to introduce the engineered fibroblasts into perfused grafts, and demonstrates the cells cause no immediate injury or functional impairment to the livers, and their engineered functions remain stable during 6 hours of perfusion. However long-term transplant studies are necessary to develop optimized protocols to ensure safety and function of biosensor cells *in vivo*. Such studies would also test the long-term fate of cells, and compare other cell types with engraftment potential into a liver, such as endothelial cells, immune cells, or even hepatocytes themselves as natural considerations. We expect that the *in vivo* durability and fate of engineered cells could vary significantly between different cell types, and it is also possible that different applications will require different cell types. Other considerations could include additional layers of safety, such as suicide switches that our team and others have developed which can also be incorporated into biosensors for simple elimination after transplantation.

Rejection of liver allografts has decreased with the use of immunosuppressive drugs, but still affects 15 to 25% of liver transplant recipients [36, 37]. Evaluation of the immunosuppression state and early diagnosis of acute or chronic organ rejection remains a crucial task, but such monitoring relies largely on clinical judgment [38]. In the case of liver, current biomarkers of transplant such as ALT and AST have poor sensitivity to detect chronic rejection; it is similarly difficult to easily assess development of graft acceptance [38]. Innate immune cytokines such as interleukins, TNF, and IFN-γ are likely first expressed in tissue before hepatic injury and may be valid detection points for biomarkers [38, 39], especially if they can be sensed locally and in tandem. We envision that this initial work can serve as a foundation for the use of engineered (synthetic) cells meant to be engrafted into an organ prior to transplant to act as an *in-situ* cell-based biosensor for potentially reporting and responding to the inflammatory state of a graft. Customizing the genetic constructs within this biosensor may also find use in dysfunctional organ donations, like fatty or cirrhotic organs, as a way to expand the donor pool itself.

In summary, a proof-of-concept study was performed to verify the engraftment and function of engineered cells perfused into an *ex vivo* organ. A feasible process of preparing an engineered cells suspension into an organ preservation solution during machine perfusion was established. This study represents the first ever demonstration of embedding engineered cells within an organ prior to transplant and can serve as the basis for new designer engineered cells for specific *in vivo* functionality along with improved perfusion and monitoring techniques to control and correlate the bioprocess to surgical outcomes post-transplantation.

## Supporting information

**S1 Fig. Bolus injection of cells fails to distribute biosensors throughout organ.** Dyed cells injected directly into the cannulated lived did not distribute as well as perfused cells.
(TIF)

## Author Contributions

**Conceptualization:** Biju Parekkadan.

**Data curation:** Ling-Yee Chin, Cailah Carroll, Siavash Raigani, Danielle M. Detelich, Shannon N. Tessier, Stephen P. Schmidt.

**Formal analysis:** Ling-Yee Chin, Cailah Carroll, Siavash Raigani, Danielle M. Detelich, Shannon N. Tessier, Gregory R. Wojtkiewicz, Heidi Yeh, Korkut Uygun, Biju Parekkadan.

**Funding acquisition:** Ralph Weissleder, Korkut Uygun, Biju Parekkadan.

**Investigation:** Ling-Yee Chin, Cailah Carroll, Siavash Raigani, Danielle M. Detelich, Shannon N. Tessier, Heidi Yeh, Korkut Uygun, Biju Parekkadan.

**Methodology:** Ling-Yee Chin, Cailah Carroll, Siavash Raigani, Danielle M. Detelich, Shannon N. Tessier, Biju Parekkadan.

**Project administration:** Ralph Weissleder, Heidi Yeh, Korkut Uygun, Biju Parekkadan.

**Resources:** Ralph Weissleder, Heidi Yeh, Korkut Uygun, Biju Parekkadan.

**Supervision:** Ralph Weissleder, Heidi Yeh, Korkut Uygun, Biju Parekkadan.

**Validation:** Ling-Yee Chin, Cailah Carroll, Siavash Raigani, Shannon N. Tessier, Ralph Weissleder, Korkut Uygun, Biju Parekkadan.

**Visualization:** Gregory R. Wojtkiewicz, Stephen P. Schmidt.

**Writing – original draft:** Ling-Yee Chin, Cailah Carroll, Siavash Raigani, Gregory R. Wojtkiewicz, Ralph Weissleder, Heidi Yeh, Korkut Uygun, Biju Parekkadan.

**Writing – review & editing:** Ling-Yee Chin, Cailah Carroll, Siavash Raigani, Ralph Weissleder, Heidi Yeh, Korkut Uygun, Biju Parekkadan.

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
