## [Decision Letter · Decision Letter 0]

5 Sep 2019

PONE-D-19-19424

Ex Vivo Perfusion-Based Engraftment of Genetically Engineered Cell Sensors into Transplantable Organs

PLOS ONE

Dear Professor Parekkadan,

Thank you for submitting your manuscript to PLOS ONE. After careful consideration, we feel that it has merit but does not fully meet PLOS ONE’s publication criteria as it currently stands. Therefore, we invite you to submit a revised version of the manuscript that addresses the points raised during the review process.

The critical concerns are raised on the preservaton of hepatic function after used the cellular biosensor sysytem (considerlly, such as two cuff method might be a pratical way to examine the hepatic function of recipient animal after liver transplantation) , and the comparison of different cell sources, fibroblasts vs mesenchymal stromal cells or mesenchymal stem cells  Please find the following constructive parts of comments of the reviewers.  

We would appreciate receiving your revised manuscript by Oct 20 2019 11:59PM. To enhance the reproducibility of your results, we recommend that if applicable you deposit your laboratory protocols in protocols.io, where a protocol can be assigned its own identifier (DOI) such that it can be cited independently in the future. For instructions see: http://journals.plos.org/plosone/s/submission-guidelines#loc-laboratory-protocols

We look forward to receiving your revised manuscript.

Kind regards,

Yun-Wen Zheng

Academic Editor

PLOS ONE

Journal Requirements:

1. Thank you for including your competing interests statement; "A patent on engineered cell technology and uses in organ transplantation has been filed. The authors declare no other competing financial interests."

We note that you have a patent relating to material pertinent to this article. Please provide an amended statement of Competing Interests to declare this patent (with details including name and number), along with any other relevant declarations relating to employment, consultancy, patents, products in development or modified products etc. Please confirm that this does not alter your adherence to all PLOS ONE policies on sharing data and materials, as detailed online in our guide for authors http://journals.plos.org/plosone/s/competing-interests by including the following statement: "This does not alter our adherence to  PLOS ONE policies on sharing data and materials.” If there are restrictions on sharing of data and/or materials, please state these. Please note that we cannot proceed with consideration of your article until this information has been declared.

2. Thank you for including the follwoing funding information within your manuscript; "This research was conducted with support under Grant No. R21AI134116 (BP, KU)

awarded by the National Institutes of Health and support from the Shriners Foundation

for Children."

"A patent on engineered cell technology and uses in organ transplantation has been filed. The authors declare no other competing financial interests."

Reviewers' comments:

Reviewer's Responses to Questions

**Comments to the Author**

1. Is the manuscript technically sound, and do the data support the conclusions?

Reviewer #1: Yes

Reviewer #2: Partly

2. Has the statistical analysis been performed appropriately and rigorously? 

Reviewer #1: Yes

Reviewer #2: Yes

3. Have the authors made all data underlying the findings in their manuscript fully available?

Reviewer #1: Yes

Reviewer #2: Yes

4. Is the manuscript presented in an intelligible fashion and written in standard English?

Reviewer #1: Yes

Reviewer #2: Yes

5. Review Comments to the Author

Reviewer #1: In the study entitled “Ex Vivo Perfusion-Based Engraftment of Genetically Engineered Cell Sensors into Transplantable Organs”, Parekkadan et al. proves an innovative concept of integrating synthetic reporter cells ex vivo into organs as a transplant-within-a-transplant during functional organ preservation with a vision to use cell biosensors as a broad way to monitor and treat tissue transplants. Overall, this work has a degree of novelty and may be used for clinical application in future. However, I have a few comments as followings:

1. The author concluded that the method of Ex Vivo Perfusion-Based Engraftment in this study could be used for therapeutic application. However, they only tested the feasibility of Ex Vivo Perfusion-Based Engraftment using fibroblast cell model. As far as we know, a body of evidence shows that stem cells can display beneficial therapeutic effects on the rejection of liver transplantation and it is not hard to isolated stem cells from allogeneic sources. Therefore, in this regard I suggest that stem cell model should be tested at least.

2. In fact, the mechanisms underlying the therapeutic effects on the rejection of liver transplantation are not fully elucidated. In many studies stem cells have been reported to function by secreting some key factors and exosomes, so it seems there may be no need for stem cells to dwell or distribute specifically in liver and then display protective effects after liver transplantation. Therefore, I suggest the authors to compare the therapeutic effects of stem cells between traditional injection method and Ex Vivo Perfusion-Based Engraftment method using rat liver transplant model.

Reviewer #2: Parekkadan, et al., present an interesting proof of concept study, wherein they demonstrate the ability of a machine-based perfusion system to distribute transducer fibroblasts, which produce Luciferase, into rat livers as demonstration that a system using in vivo biosensors might someday be used in the solid organ transplant settings to detect clinical events.

The overall concept is innovative and “out of the box.” They demonstrate in this article the ability to take fibroblasts, transduced with lentiviral vectors, and demonstrate that they successfully distribute throughout these organs. However, a number of questions that are relevant to the demonstration of proof of concept and practical ability to stimulate future translation, are worth considering.

First, one critical piece of this is demonstration that the distributed cells do not impair hepatic function. It is not clear to me that the demonstrations here are sufficient. For example, there is minimal edema and AST levels did not rise significantly, but the ideal demonstration of this would have been transplantation of a cellular biosensor-infused liver into an allogeneic (or even syngeneic) recipient with demonstration that the recipient animal can sustain adequate hepatic function.

Second, the choice of fibroblasts (vs., as discussed frequently, mesenchymal stromal cells that have a relatively broad history in clinical transplant settings) might become problematic, with the potential to induce inflammation vs. other cell types. Also the choice of a vector system capable of propagating into daughter cells might raise concerns about inflammatory (or even neoplastic) potential in the real world setting.

All of the above concerns do not undermine the interesting concept, but they raise issues about what level of proof of concept demonstration (e.g., if not sufficient to determine lack of impact on hepatic function and using different cells or vectors than might realistically be used in initial clinical application) has been achieved herein.

6. PLOS authors have the option to publish the peer review history of their article (what does this mean?). If published, this will include your full peer review and any attached files.

Reviewer #1: No

Reviewer #2: No

---

## [Author Response · Author response to Decision Letter 0]

24 Oct 2019

See attached submission for responses to reviewer's comments

---

## [Editor Report · Decision Letter 1]

31 Oct 2019

Ex Vivo Perfusion-Based Engraftment of Genetically Engineered Cell Sensors into Transplantable Organs

PONE-D-19-19424R1

Dear Dr. Parekkadan,

We are pleased to inform you that your manuscript has been judged scientifically suitable for publication and will be formally accepted for publication once it complies with all outstanding technical requirements.

With kind regards,

Yun-Wen Zheng

Academic Editor

PLOS ONE
---

## [Editor Report · Acceptance letter]

22 Nov 2019

PONE-D-19-19424R1 

Ex vivo perfusion-based engraftment of genetically engineered cell sensors into transplantable organs 

Dear Dr. Parekkadan:

I am pleased to inform you that your manuscript has been deemed suitable for publication in PLOS ONE. Congratulations! Your manuscript is now with our production department. 

With kind regards,

on behalf of

Dr. Yun-Wen Zheng 

Academic Editor

PLOS ONE